# Laser Direct Joining of Steel to Polymethylmethacrylate: The Influence of Process Parameters and Surface Mechanical Pre-Treatment on the Joint Strength and Quality

**DOI:** 10.3390/ma15145081

**Published:** 2022-07-21

**Authors:** Fábio A. O. Fernandes, José P. Pinto, Bruno Vilarinho, António B. Pereira

**Affiliations:** 1TEMA—Centre for Mechanical Technology and Automation, Department of Mechanical Engineering, University of Aveiro, Campus de Santiago, 3810-193 Aveiro, Portugal; josemcp@ua.pt (J.P.P.); b.vilarinho@ua.pt (B.V.); abastos@ua.pt (A.B.P.); 2LASI—Intelligent Systems Associate Laboratory, Portugal

**Keywords:** laser welding, dissimilar joining, metal-polymer joining, PMMA, laser conduction joining, surface mechanical pre-treatment

## Abstract

The search for lightweight structures increases the demand for non-metallic materials, such as polymers, composites, and hybrid structures. This work presents the dissimilar joining through direct laser joining between polymethylmethacrylate (PMMA) and S235 galvanised steel using a pulsed Nd:YAG laser. The main goal is to determine the influence of processing parameters on joint strength and quality. In addition, the impact of surface conditions on the joint quality was also analysed. Overall, the optimum ranges of process parameters were found, and some are worth highlighting, such as the laser beam diameter and pulse duration, which significantly influenced the joint strength. Failure of the welded samples occurred in PMMA component, demonstrating good joint efficiency. Additionally, a maximum increase of 5.1% of the tensile shear strength was achieved thanks to the mechanical pre-treatment. It is possible to conclude that the joining between PMMA and the S235 galvanised steel can be performed by optimising the process parameters. Additionally, it can be enhanced through surface pre-treatments by exploring the mechanical interlock between both materials.

## 1. Introduction

Thermoplastic and metallic materials have several applications in many relevant industries. For instance, structural applications for the automotive and aerospace industries or even biomedical devices [1,2,3]. Joining dissimilar materials is highly attractive due to the expected properties of hybrid components and the flexibility in product design [4]. The potential cost reduction and functional integration for large series products make it also interesting for the electronics and household appliance industries [5].

The joining of dissimilar materials, particularly between polymeric and metallic components, can be achieved through adhesive bonding or mechanical fasteners. Nevertheless, new joining technologies such as laser-based ones allow polymers and metallic alloys without adhesives, fillers, or additional fasteners. Laser-based processes make it possible to reduce processing times, eliminate curing stages, and increase flexibility and long-term stability by employing the laser, among many other advantages such as the non-contact energy input. These are exceptionally significant at the industrial level [6]. One example is laser-assisted metal and plastic (LAMP) joining, a direct laser joining technique for metal-plastic bonding, which has been reported to form strong joints [7,8,9,10].

The LAMP or direct laser joining principle is based on the heating of the metal-plastic interface thanks to the direct incidence of the laser beam. There are two variants of the process, depending on the material that the laser beam targets, always having the metallic part as the radiation absorber component and having both materials in contact in an overlap configuration. Thus, the process can be carried out by transmitting the laser through the thermoplastic and heating directly the interface. Or, by having the laser directly on the metallic component, increasing its temperature, and through thermal conduction, it will heat the interface with the plastic part, causing its melting.

Heat conduction joining is mainly used in the laser-based joining of metals with polymers [5]. Recently, Schricker et al. (2020) addressed the energy efficiency of the process based on dissipated heat by analysing the influence of process parameters (focal diameter, joining speed, energy per unit length), metallic materials (AA 6082, AISI 304), geometric parameters (overlap width, material thickness) and various polymers such as polypropylene (PP), polyamide (PA) 6 and PA 6.6.

Poor melting at the interface results in a weak joint, demonstrating the importance of determining the optimum laser parameters. On the other hand, excessive energy results in polymer decomposition, leading to the formation of bubbles, which will be entrapped on the metal-plastic interface after the solidification of the melted thermoplastic. After re-solidification, mechanical interlocking at the surface, chemical reactions, and Van Der Waals’ interaction occur. These are the mechanisms responsible for stable bonds at the interface between plastic and metal [2,11,12]. Although bubbles formation is inevitable, it can be reduced by optimising the laser parameters. Chen et al. [4] developed a laser joining technology with ultrasonic vibration and using a 300 W Nd-YAG pulsed laser to reduce bubble formation at the interface of laser joining of polyethylene terephthalate (PET) and titanium. Laser power, scanning speed, material transmissibility, clamping pressure and absorption properties at the interface are essential parameters for successful welds between metals and polymers [13].

In addition to the laser processing parameters and process configuration, the effects of surface physical pre-treatment of both metallic and plastic parts on the joint strength have been also studied [9,14,15,16]. The molten material wets the metal surface and fills the surface structures, which then cools and solidifies. Therefore, the surface of the metallic component plays an essential role in the mechanical interlocking and the physicochemical interactions between polymer and metal.

Due to the different applications, several materials have been studied as joining elements for metal-polymer hybrid composites: metals-stainless steels [5,17], titanium alloys [4], magnesium [18] or aluminium alloys [5,16]; thermoplastics–polycarbonate (PC) [17], PA [5,19], PET [4], or PP [5,20]. The literature has also explored the polymer-metal hybrid joining of polymers with aggregates or reinforced with other materials. Heckert et al. [16] studied the influence of laser surface pre-treatments on aluminium joints (with glass fibre reinforced thermoplastics). More recently, Rodríguez-Vidal et al. [21] performed surface modification pre-treatment on direct laser joining of glass fibre reinforced PA to a low-alloy steel HC420 to investigate the effect of metal micro-grooving on the joint mechanical strength.

A few attempts have been made to join polymethylmethacrylate (PMMA) with metals. The welding of PMMA to steel alloys would make it possible to employ this joining technique in various products, eliminating the need for adhesives, mechanical fasteners, or fillers. Laser welding is a more flexible joining process without adhesives, which often significantly impact the environment. Several potential applications include display cases, medical and laboratory equipment, protective cages for industrial equipment, appliances, kitchenware, etc. In other words, any application where there is the need for transparent components where PMMA properties are suitable and substituting the adhesive bonding or mechanically fasteners between PMMA and steel structures.

In one rare example, Hussein et al. [22] reported the welding between transparent PMMA and stainless steel 304 sheets using a pulsed mode Nd:YAG laser by conduction joining. The peak power, pulse duration, pulse repetition rate, scanning speed, and pulse shape were reported to significantly affect the joint strength, joint width, and joint quality and appearance. Recently, Huang et al. [23] also addressed the dissimilar joining of 304 stainless steel to PMMA with an Nd:YAG pulsed laser, optimising the weld strength based on the process parameters. It was concluded that the laser pulse duration has the maximum effect on weld strength, followed by laser peak power and welding speed. In contrast, pulse frequency had an insignificant effect on it.

Although scarce in the literature, the joining of PMMA with structural steel has also been investigated. In one rare example, Bauernhuber et al. [24] studied the joining between PMMA and the S235 (unalloyed) cold drawn structural steel. The observations were focused on the bubble formation and not on the weld strength and its optimisation based on the process parameters. Csiszér et al. [25] investigated the joining between cellulose fibre-reinforced PMMA and S235JR structural steel, using a solid-state Yb:YAG laser to find the adequate wavelength and optimal laser parameters.

This work aims to find the optimal parameters to join PMMA and S235 galvanised steel. The dissimilar joining is carried out with a pulsed Nd:YAG laser through conduction joining. The quality of the joints is assessed by performing tensile tests to determine mechanical strength. These are carried out to induce shear failure in the joint between PMMA and steel. Additionally, another goal of this work is to analyse the effect of the surface mechanical pre-treatment on the quality of the joint. Therefore, in the second stage of this work, the metallic samples were subjected to grinding prior to the laser conduction joining process and then submitted to mechanical testing.

## 2. Materials and Methods

### 2.1. Materials

In this work, the thermoplastic used was PMMA, supplied by Plexicril in 3 mm thick sheets. These were cut on a band saw to obtain 40 × 14 × 3 mm samples. Then, these were polished to guarantee the necessary surface finishing and eliminate any imperfection introduced by the cutting process. Table 1 presents the relevant properties of PMMA for laser welding.

Regarding the metallic part, 0.8 mm thick S235 galvanised steel sheets were cut in a vertical guillotine to obtain 40 × 14 × 0.8 mm samples. Sample thickness was selected based upon the findings of a preliminary study. Table 2 presents the relevant properties of EN 10025-2 S235 steel for laser welding. Both Table 1 and Table 2 present properties based on the information provided by the suppliers. Additional tensile tests were performed on PMMA according to ISO 527-1 [26], obtaining an average tensile strength of 48 MPa. Figure 1 shows an illustration of the lap-joint configuration and the specimen dimensions.

### 2.2. Laser and Processing Parameters

One of the goals is to determine the optimal laser welding parameters. A pulsed Nd:YAG laser was used to perform overlap joints (Figure 2). The samples were welded in the Sisma SWA300 Nd:YAG laser machine, using a fundamental wavelength of 1064 nm. It has an average maximum power of 300 W and a peak pulse power of 12 kW. As depicted in Figure 2, samples were welded according to the direct laser joining principle through heat conduction since the metallic piece absorbs the radiation from the laser, generating heat responsible for the melting of the PMMA.

Intimate contact between the parts to weld is necessary to achieve successful joining. Therefore, a clamp system was developed with two M4 screws attached to each cover plate (Figure 3d). This apparatus guarantees the contact between both parts to weld in a lap configuration. The four-screw clamping system was set to apply a pressure of approximately 0.8 MPa. Figure 3 depicts the experimental setup.

The parameters selected for optimization, based upon the experimental conditions and limitations of the equipment employed, were laser power (percentage of the peak power), pulse duration, percentage of overlapping between pulses and laser beam diameter. The number of beads was also considered as one of the parameters. Single passes were performed as a first trial, but weak joints were achieved. Therefore, a minimum of two beads was defined. In this work, one scanned line is defined as a bead, and the following weld passes are performed next to the previous one (Figure 4). In other words, the number of weld beads corresponds to the number of weld passes.

Table 3 displays the 20 discrete parameter sets employed in the study. The definition and evolution of these was based upon sequential observations, varying one parameter and fixing the others. The lower and upper limits of the investigated parameters were defined based on preliminary experiments. A constant 8 mm/s scanning speed of the laser beam onto the steel surface was employed in all the experiments. After each set, tensile tests were carried out to adjust the parameters progressively.

### 2.3. Surface Mechanical Pre-Treatment

In addition to the search for the optimal welding parameters, the influence of the surface conditions was also analysed. The steel pieces were subjected to a surface pre-treatment on the overlapping contact side. This mechanical surface pre-treatment is grinding. A series of sandpapers obtained from Indasa—100, 240 and 600 grit—were employed to remove pre-existing surface oxides and ensure consistent surface roughness. Different grits will cause different levels of abrasion, which will make it possible to assess its impact on the joining process. The mechanical grinding treatment was performed in a Struers RotoPol-21 machine to guarantee a uniform surface treatment.

Each steel sample was ground for 5 min. After grinding, the pieces were cleaned with soap and water for 10 min and, in the end, rubbed with ethanol and dried, finishing the surface physical pre-treatment process. The samples subjected to surface pre-treatment were then welded with the best set of parameters from Table 3, identified after the primary mechanical testing stage–samples welded without surface physical pre-treatment. Figure 4 presents the samples after the surface mechanical pre-treatment and prior-welding and post-welding.

### 2.4. Tensile Testing

In order to assess the quality of the lap joints and determine their mechanical strength, the welded samples were submitted to tensile testing, which will induce shear in the joints. The mechanical tests were performed in the Shimadzu AGX 10 kN universal testing machine at 1 mm/s. All the samples welded with one of the 20 sets of parameters, independently of the surface physical pre-treatment or its absence, were subjected to tensile loading to assess the joining quality between PMMA and the S235 galvanised steel. Figure 5 shows one welded sample being tested. Since these are overlap joints, two parts of the same materials and equivalent thicknesses were coupled with the samples in the fixation, aligning the neutral axis with the weld zone, and thus, minimising the bending moment and guaranteeing essentially shear stresses in the joint. Naturally, there will always be tensile stresses due to the remaining moment. Nevertheless, these are greatly minimised with the adopted configuration.

## 3. Results

### 3.1. Optimal Laser Parameters

The first testing stage corresponds to the welds between PMMA and steel without surface physical pre-treatment other than the galvanising of the steel parts. Figure 6 presents some welded specimens labelled with the corresponding set of parameters. At the same time, Table 4 shows the results of the tensile tests, including the mechanical strength and strain at failure. In addition to the tensile tests, the quality of the joints was also assessed by visual inspection, particularly the observation of bubbles formation and material decomposition, which typically correlate low strength joints between PMMA and steel. Hussein et al. [22] reported weaker joints between PMMA and stainless steel, mainly due to polymer decomposition, bubble formation and long cavity formation along the joint, identifying roasting, charring, vaporisation and degradation of the PMMA at the joint interface. The ideal welded joint would have the same strength as an adhesive if aiming for a specific application, or the same strength of the weaker base material, in this case, PMMA. However, based on the literature, it would be an excellent result to achieve a welded joint with half the tensile strength of the base material.

The shear strength was calculated by dividing the maximum force by the joining area when the joint failed. This area corresponds to the spot diameter multiplied by the length of the weld. Nevertheless, stronger joints were achieved to the point that failure occurred in the PMMA part and not the joint. In these particular cases, the cross-sectional area of the PMMA part was used to calculate the stress. In addition, failure was always achieved at the end of the metallic part. During the welding process, this extremity penetrated the PMMA sample (around 1 mm on average), causing a reduction in the cross-section of PMMA and originating stress concentrations in this region. Therefore, the cross-sectional area of the PMMA part (Figure 1) was updated accordingly for the calculations. Additionally, Figure 7 shows the typical stress-strain curve obtained for this type of joint. This particular one refers to set 14.

The joint strength of samples welded with the parameters sets 1, 4, 8, 13, 15, and 19 are not presented because these were not correctly joined. Starting with the peak power, within the defined range, it is possible to conclude that its optimal value is around 68–70%. Lower power is not enough, resulting in weaker joints. However, higher peak power is excessive for PMMA, resulting in its decomposition, and thus, poor joining and aesthetics. Such values lead to bubbles formation within the interface polymer-metal, and in extreme cases, to burnt material.

Regarding the beam diameter, its influence on the joint performance is also significant. A laser beam with a diameter of 2 mm resulted in superior joints than the sets with more concentrated laser beams. The laser beam effect on the quality of the joints between PMMA and the S235 galvanised steel is noteworthy. Additionally, it is also necessary to mention the limitations of the laser equipment employed in this study, i.e., the upper limit of the laser beam diameter is 2 mm. Based on this finding, it is plausible to infer the potential improvement of the polymer-metal joint if wider laser beams could be employed. It is also important to mention the exceptional case of set 12, which presented a positive result although carried out with a 1.5 mm laser beam.

Regarding the overlapping between beams, based on the results presented in Table 4 and on the observations during the welding process, it became evident that pulse overlapping inferior to 75% is insufficient to join both materials within the defined peak power range. On the other hand, values equal to or higher than 90% result in excessive heat, leading to bubbles and polymer decomposition. This range of pulse overlapping demonstrates the PMMA sensitivity to the amount and rate of energy supplied to the joint, and thus, the maximum temperature reached and the heating rate. Good results were achieved within the 80–85% overlapping (with the best results close to the 85%). Contrary to the other parameters, and based on the defined range, the pulse duration variation did not significantly improve, and no trend was observed.

The influence of weld beads or passes was also analysed: 2, 3 and 4 bead joints. The latter resulted in poor results, and thus, an upper limit was found for the number of beads. Therefore, the minimum and maximum limits are 2 and 3 weld beads, respectively. These correspond to the number of welded passes. It is worth highlighting the results achieved with two beads, which, in most scenarios, resulted in higher strength weld joints when compared to the cases with three beads. Again, as discussed previously, the way the energy is transferred to a specific area and volume of material influences the melting of PMMA, mainly because of the temperatures reached and the heating rate, which explains why there is an optimum range for each parameter.

Figure 8 summarises the evolution of the joint strength with the process parameters. In this case, 10 3D graphs depict the interaction between process parameters and their effect on the joint strength. This analysis shows the optimum combination and ranges for each parameter, highlighting some previously discussed findings based on Table 3 and Table 4. For instance, Figure 8c demonstrates that the joint weld strength is inversely proportional to the number of beads for any percentage of overlapping. Nevertheless, the best weld strengths were achieved for overlapping of 85%. Similar conclusions can be made for other interactions with the number of beads and laser beam diameter, for instance, with the peak power (Figure 8e and Figure 8f, respectively). In addition, the pulse duration interaction with the laser beam diameter shows a similar evolution, although not proportional for larger beams (Figure 8j).

Figure 8d shows a complex influence of the interaction between pulse duration and overlapping, reaching, for instance, minimum and maximum values for the same overlapping (e.g., at 85%). The same can be said about the interactions of peak power and pulse duration (Figure 8h) and between the number of beads and pulse duration (Figure 8i). Nevertheless, in both cases, the pulse duration had a significant role in improving the quality of the joint. As already referred, the energy is conducted to the interface between both materials. Thus, the heating rate and maximum temperature reached at this region must be within an optimum range to obtain good quality welds. In most cases, joints with superior strength were achieved by increasing the pulse duration.

Overall, three sets of parameters stand out positively–sets 12, 14 and 18. This conclusion is not solely based on the strength values presented in Table 4 but also on the ruptured region. For samples welded with sets 14 and 18, failure occurred in the PMMA part and not in the joint. On the other hand, set 12 failed in the joint. Figure 9 shows some specimens welded with parameter sets 14 and 18, depicting the failure region in the PMMA and presenting a preserved joint.

The strength evolution was not proportional or linear with the number of laser passes. Best results were achieved with two passes. In addition, the laser peak power, laser beam diameter and the percentage of overlapping between beams were some of the most influential parameters on the joint strength.

Although the tensile strengths obtained in welded samples were below 48 MPa, the failure during the tensile testing was not in the joint but rather in the PMMA part. Therefore, this failure below the strength of the base material may indicate a stress concentration caused by the metal component during the tensile test that, in some cases, can reduce the material failure by almost 50%. Nevertheless, the joint strength achieved was 49.8% of the strength of the base material, with failure outside the joint region, indicating that good quality welds were performed.

### 3.2. Surface Mechanical Pre-Treatment

Three sets of parameters were identified as the ones providing better results. The S235 galvanised steel samples were subjected to an abrasive surface physical pre-treatment with three different grits. Then, for each grit, these samples were welded with PMMA according to three sets of parameters identified in the previous section—12, 14 and 18. Table 5 presents joint strengths for samples welded with sets 12, 14 and 18 after surface mechanical pre-treatment of the S235 galvanised steel.

The samples subjected to surface mechanical pre-treatment with 100 grit sandpaper and welded with the parameters set 12 resulted in a weak joint that failed while being handled before tensile testing. Overall, both coarser and finer sandpapers independently of the set resulted in weaker joints than the results obtained with samples abraded with 240 grit sandpaper, which provided a superior mechanical interlock between both materials. Figure 10 depicts the better influence of the 240 grit sandpaper on the joint strength for three sets of laser welding parameters.

Therefore, it is concluded that 240 is the optimal value for the sandpaper grit to be employed in the surface mechanical pre-treatment. Nevertheless, the effect of surface physical pre-treatment was negative for set 12. On the other hand, although positive for sets 14 and 18, the increase achieved in the joint strength was negligible for set 14 and minimal for set 18. Nevertheless, on average, a rise of 5.1% was achieved in the latter with the surface mechanical pre-treatment. The maximum strength of 25.11 MPa can be justified by the already highlighted indentation caused by the metallic component during welding, reducing the cross-sectional area of the PMMA, and the stresses caused by the always inevitable bending moment in lap joints.

Overall, with set 18 for samples abraded with 240 grit sandpaper, a tensile strength of 52.3% of the base material (PMMA) was achieved. Although still far from the tensile strength of the base material, it is worth highlighting that the failure was not in the joint but actually in the PMMA part. Two factors play significant roles in this result. First, the indentation in the PMMA part is caused by the metallic one during welding, reducing its cross-sectional area and creating a region prone to stress concentration, thus, failure. Secondly, lap joint configurations are also prone to bending moment that negatively affects strength. Although preventive measures were taken to minimise it, the presence of residual stresses is inevitable.

Surface physical pre-treatments can improve the welds between PMMA and the S235 galvanised steel, as reported for other materials in direct laser joining through heat conduction between polymers and metals. In this work, the same was concluded, even though the joint strength increased slightly higher than 5%.

## 4. Conclusions

Lap welded joints were performed between PMMA and S235 steel with an Nd:YAG laser through conduction joining. An optimum range for the laser process parameters was discovered, directly connected to the energy transferred to the interface and its rate. In other words, these optimum ranges for each parameter define, as a set, the maximum temperature and the heating rate at the interface, which must be within the PMMA limits to avoid decomposition of the polymer and bubbles formation, promoting a suitable distribution of heat. Although the tensile strength obtained in welded samples was below the strength of the base material, the best joints failed in the PMMA part and not the joint. This failure below the strength of the base material is justified by the indentation caused by the metallic component during welding, reducing the cross-sectional area of the PMMA and the stresses caused by the always inevitable bending moment in lap joints. Nevertheless, the higher strength achieved was superior to 50 % of the strength of the base material, with failure outside the joint region, indicating that excellent welds were achieved.

The surface physical pre-treatments, i showed how to improve the direct laser joining through heat conduction between PMMA and the S235 galvanised steel thanks to the mechanical interlock. Nevertheless, although a 5.1% increase in joint strength was achieved, only three types of grits were explored.

Literature regarding the joining between PMMA and metallic components is scarce. The welding of PMMA to steel would make it possible to employ this joining technique in a wide variety of products, eliminating the need for adhesives, mechanical fasteners, or fillers with benefits for the environment and manufacturers with a more flexible solution. This preliminary work demonstrates the joining of PMMA, a material widely used in the automotive industry, with structural steels such as the S235, widely employed in many day-to-day applications, not only structural but also in the automotive and across the myriad industrial manufacturing sectors.

## Figures and Tables

**Figure 1 materials-15-05081-f001:**
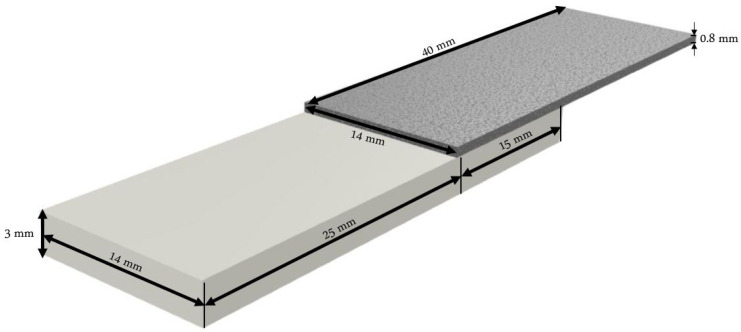
Lap-joint configuration: specimen geometry and dimensions. Thicker specimen represents PMMA, while the thinner one is steel.

**Figure 2 materials-15-05081-f002:**
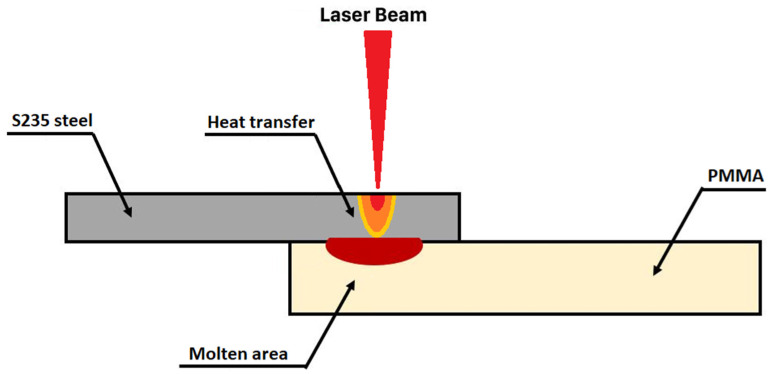
Schematics of the laser direct joining technique through heat conduction.

**Figure 3 materials-15-05081-f003:**
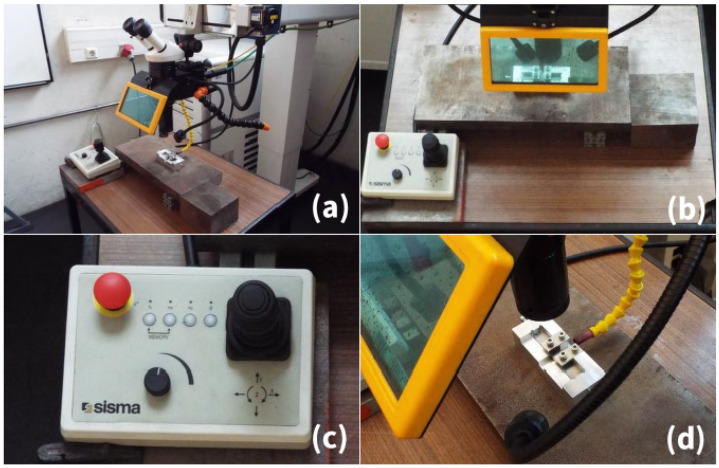
Experimental setup: (**a**) overview of the experimental setup; (**b**) shielding; (**c**) control; (**d**) clamping system.

**Figure 4 materials-15-05081-f004:**
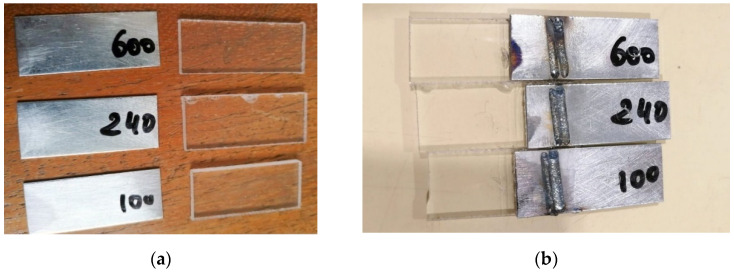
Samples after the surface physical pre-treatment: (**a**) prior welding; (**b**) post-welding.

**Figure 5 materials-15-05081-f005:**
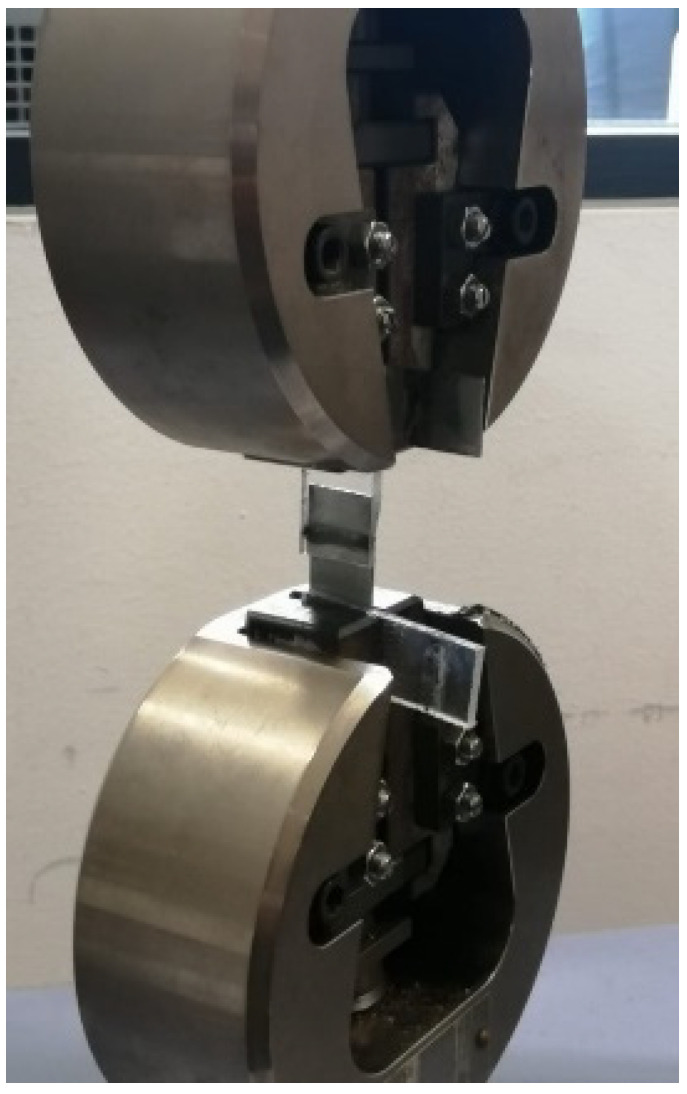
Tensile testing of the welded samples: tensile-shear configuration.

**Figure 6 materials-15-05081-f006:**
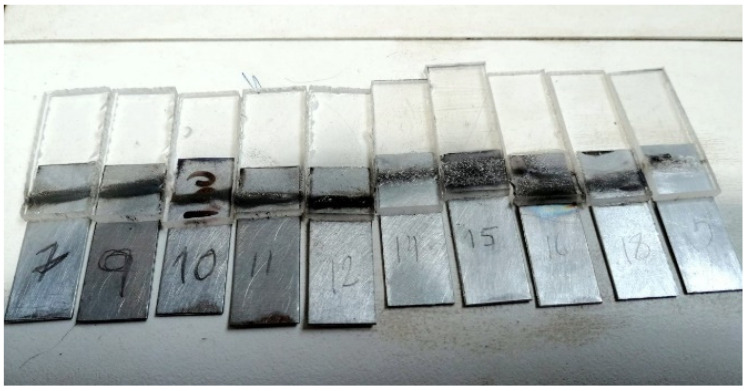
PMMA-steel (without surface physical pre-treatment) welded specimens labelled with the corresponding set of parameters.

**Figure 7 materials-15-05081-f007:**
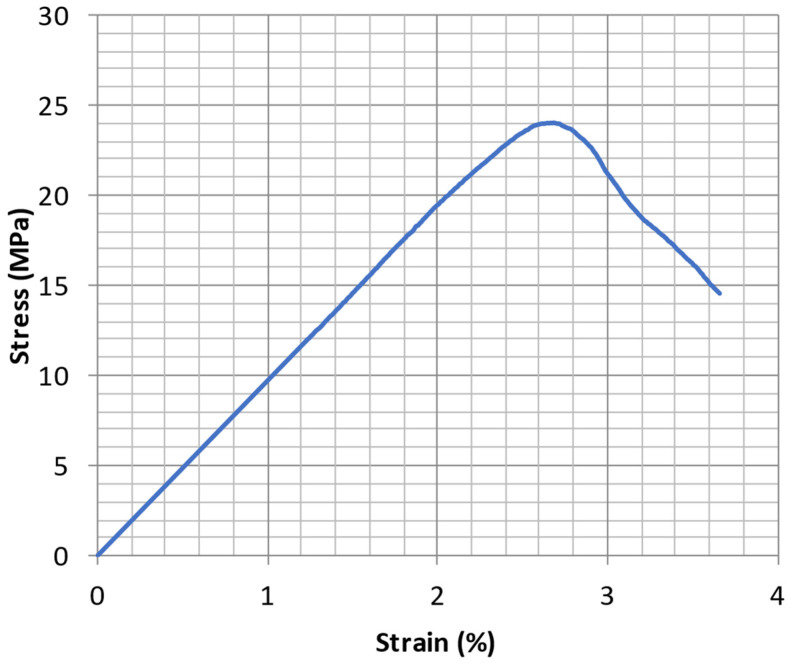
Stress-strain curve of PMMA-steel joint obtained for set 14.

**Figure 8 materials-15-05081-f008:**
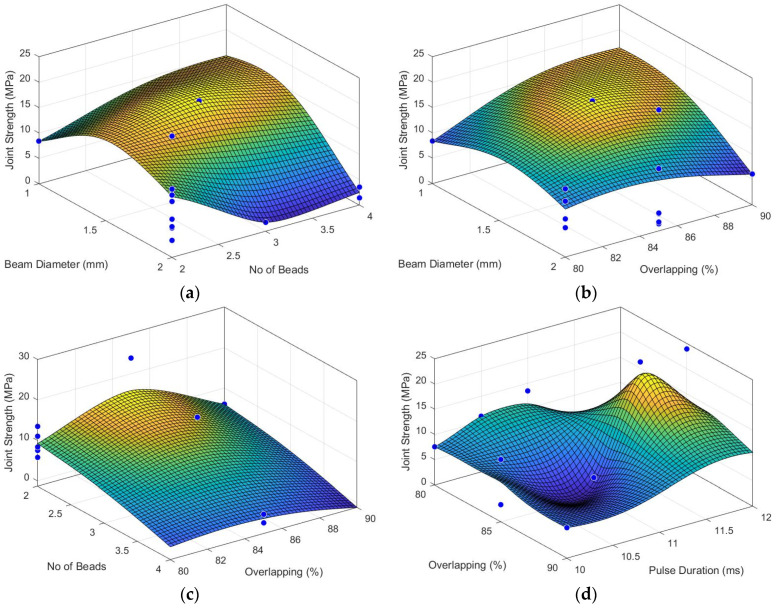
Joint strength evolution with the process parameters (blue dots represent results obtained for each set): (**a**) laser beam diameter and the number of beads; (**b**) laser beam diameter and percentage of overlapping; (**c**) the number of beads and percentage of overlapping; (**d**) percentage of overlapping and pulse duration; (**e**) peak power and the number of beads; (**f**) peak power and laser beam diameter; (**g**) peak power and overlapping; (**h**) peak power and pulse duration; (**i**) number of beads and pulse duration; (**j**) laser beam diameter and pulse duration.

**Figure 9 materials-15-05081-f009:**
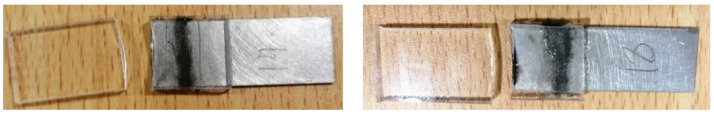
Failure of the specimens welded with the parameters sets 14 (**left**) and 18 (**right**)—failure in the PMMA part after tensile testing.

**Figure 10 materials-15-05081-f010:**
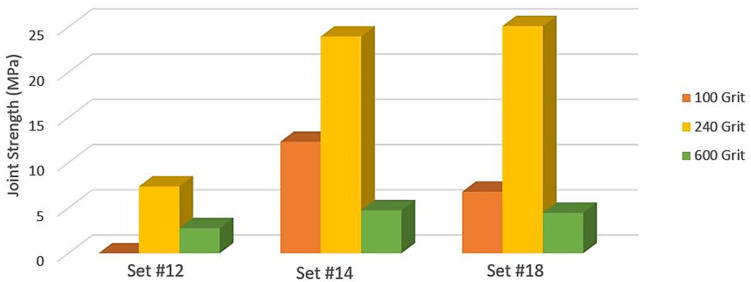
The influence of the size of abrasive materials employed on the surface mechanical pre-treatment on the joint strength for three sets of laser welding parameters.

**Table 1 materials-15-05081-t001:** Physical and mechanical properties of the PMMA for laser welding.

Property	Value
Density (kg/m^3^)	1190
Heat Capacity (J/K)	1.32
Specific Heat Capacity (J/kg·K)	1466
Thermal Conductivity (W·m^−1^·°C^−1^)	0.19
Crystallinity (%)	5–15
Melting Temperature (°C)	180
Tensile Strength (MPa)	48

**Table 2 materials-15-05081-t002:** Physical and mechanical properties of the S235 for laser welding.

Property	Value
Density (kg/m^3^)	7850
Specific Heat Capacity (J/kg·K)	460–480
Thermal Conductivity (W·m^−1^·°C^−1^)	40–45
Melting Temperature (°C)	1480–1526
Tensile Strength (MPa)	360–510

**Table 3 materials-15-05081-t003:** Welding parameters.

# Set	Peak Power (%)	Pulse Duration (ms)	Pulse Overlapping (%)	Beam Diameter (mm)	No. of Beads/Passes	Energy (J)
1	68	10.0	90	2.0	2	60.8
2	68	10.0	85	2.0	2	60.8
3	68	10.0	80	2.0	2	60.8
4	75	10.0	80	2.0	2	68.0
5	75	10.0	85	2.0	2	68.0
6	75	10.0	90	2.0	2	68.0
7	75	10.5	80	2.0	2	63.8
8	75	11.0	80	2.0	2	63.8
9	75	11.5	80	2.0	2	63.8
10	68	11.0	80	2.0	2	66.8
11	68	11.0	80	1.0	2	66.8
12	70	11.5	85	1.5	2	72.2
13	70	11.5	85	2.0	3	72.2
14	70	11.5	85	2.0	2	72.2
15	70	11.0	85	2.0	4	72.2
16	70	11.0	85	2.0	3	72.2
17	70	11.0	85	2.0	4	72.2
18	70	12.0	85	2.0	2	72.2
19	70	12.0	85	2.0	3	72.2
20	70	12.0	85	2.0	4	72.2

**Table 4 materials-15-05081-t004:** Results from the tensile tests.

# Set	ε_failure_ (%)	Joint Strength (MPa)
1	-	-
2	1.13	12.30
3	0.875	7.55
4	-	-
5	0.36	3.38
6	0.62	6.08
7	1.59	11.04
8	-	-
9	0.48	5.82
10	0.28	13.47
11	1.93	8.38
12	1.93	18.37
13	-	-
14	2.68	23.90
15	-	-
16	0.26	1.77
17	0.77	3.54
18	1.82	23.89
19	-	-
20	0.22	1.42

**Table 5 materials-15-05081-t005:** Results from the tensile tests of samples welded with sets 12, 14 and 18—S235 galvanized steel subjected to surface mechanical pre-treatment before the welding process.

# Set	Grit	ε_failure_	Joint Strength (MPa)
12	100	-	-
12	240	0.73	7.39
12	600	0.28	2.79
14	100	1.33	12.33
14	240	2.45	23.96
14	600	0.45	4.75
18	100	0.67	6.80
18	240	2.64	25.11
18	600	0.745	4.47

## Data Availability

Data is contained within the article.

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
