# Peer review of "Laser Direct Joining of Steel to Polymethylmethacrylate: The Influence of Process Parameters and Surface Mechanical Pre-Treatment on the Joint Strength and Quality"

_materials, 2022, doi:10.3390/ma15145081_

Round 1
Reviewer 1 Report
1. The specific heat capacity units in Tables 1 and 2 should be in the same standard format: J/(kg·K); in addition, the width of the second column in Tables 1 and 2 should be increased.
2. The unit of strain in Table 4 should be indicated as "%". Since the measured strength of the welded joints in this experiment is low, the accuracy of the area value used to calculate the strength has a great influence on the experimental results. The authors should describe in detail how to measure the area value accurately.
3. Mechanical properties are one of the most concerned issues in this paper, so the corresponding data should have sufficient evidence. The stress-strain curves corresponding to the test results in Tables 4-5 should be given in the manuscript.
4. The authors claim that bubbles can form on the solder interface under certain conditions. Authors should provide clear photographs as evidence. Also, what is the microstructure of the weld crack? The author should give a professional description.
5. What equation is used to draw the three-dimensional surface in Figure 7? What is the theoretical basis? These should be described in detail.
6. For Figure 8, why is the fracture always outside the welded joint? Figure 5 shows the authors using a PMMA spacer to aid in clamping. Did the sample not acquire sufficient concentricity during the tensile test?
7. The author believes that one of the reasons for the fracture of the welded joint is that the steel piece is pressed into the PMMA part and forms some kind of defect. However, the authors did not provide any direct evidence. Therefore, macroscopic and microstructural characterization of welded joints must be provided to improve the readability of this work.
8. The conclusion is somewhat lengthy and the authors are advised to shorten it. The conclusion should include a brief description of the main experimental results and a statement of their innovativeness and research significance.
9. It can be seen from Figure 2 that the laser beam is injected from the side of the steel piece, and the welding interface is not directly heated. How is the weld quality if the laser beam is injected from the side of the PMMA sample?
10. How is the welding interface formed? Is there an interfacial reaction between the steel plate and PMMA? What kind of interface is formed? What are the advantages of laser welding compared to using adhesives (such as AB glue, bond strength ~30MPa)?
Reviewer 2 Report
The paper "Laser direct joining of steel to polymethylmethacrylate: the influence of process parameters and surface pre-treatment on the joint strength and quality" by F.A.O. Fernandes, J.P. Pinto, B. Vilarinho, C.L. Alves, and A.B. Pereira is a very good work on the influence of processing parameters on joint strength and quality of direct laser joining between polymethylmethacrylate (PMMA) and S235 galvanised steel using a pulsed Nd:YAG laser.
The need for these structures is high demanding and this work has been carried out with careful attention.
The introduction includes the most important results related to the subject.
The matherial and method section is very clear and well detailed. The results as presented are convincing and well justified.
I recommend the publication in this journal.
The need for these structures is high demanding and the work has been carried out with careful attention. The topic is moderately innovative and original, but as the literature does not have many studies in this subject it is important to have this contribute.
The paper is clearly written, the introduction includes the most important results related to the subject.
The material and method section is well detailed. The resuls presented are convincing and well justified. Figures and table are clear and comprehensive.
The conclusions are consistent with the data presented.
The work is a preliminary study but already demonstrates the possibility of joining PMMA with structural steels such, and the way to enhance the joining between both through process parameters optimisation and surface pre-treatment.
I found the paper suitable for the publication in this journal.
Reviewer 3 Report
Laser direct joining of steel to polymethylmethacrylate: the influence of process parameters and surface pre-treatment on the joint strength and quality
F. A. O. Fernandes, et al.
Summary
This paper examines the properties of PMMA sample bonded to S235 Galvanised steel using a pulsed Nd:YAG laser.
As mentioned in the introduction, much previous work has tended to focus on stainless steel as a substrate, most likely with medical applications in mind. This work is significant because S235 and other grades of both mild and galvanised steel are widely employed in a huge number of day-to-day applications, not only structural, but also in the automotive and across the myriad industrial manufacturing sectors.
The work undertaken here is well-described and the results presented with clarity. The written English is particularly good, apart from the occasional typo and a few minor grammatical errors (see below). The authors are to be commended for the presentation of their manuscript.
Minor typos etc
Line 89 – impact should be singular here, not plural as written.
Line 129 – should read ‘Sample thickness was selected based upon the findings of a preliminary study.
Lines 162-63 – should read ‘The parameters selected for optimization, based upon the experimental conditions and limitations of the equipment employed, were laser power…..etc etc.
Lines 170-71 – should read ‘Table 3 displays the 20 discrete parameter sets employed in the study. The definition and evolution of these was based upon sequential observations…..etc…etc….
Line 278 – in the brackets it should read (with the best results close to 85%)
Figure 7 – I think Figure 7 should have the y-axis the same in all the digrams (i.e. 0-30) – I was constantly needing to check the axis values.
Figure 9 – is this correct ? Table 4 says the maximum for set#12 was 18.37 (MPa) – it is represented in Figure 9 as around 5 MPa.
Line 371 – it should read ‘abraded’ not ‘abrased’
General comments and suggestions
Line 128 – please check your S235 sample, is it really galvanised ? or is it mild steel, please check.
When characterising the samples I think it would have been good to include either some optical microscopy or SEM images of the surfaces before and after preparation, so the reader can judge the effect of the sandpapers.
If not microscopy, then maybe a profile of surface roughness by ‘Talysurf’ or even Atomic Force Microscopy would show the differences. However, maybe these options can be considered in the next paper.
One other thing that bothers me is the constant use of the phrase ‘pre-treatment’. In metal finishing, the term pre-treatment is synonymous with chemical pre-treatment using chromate or, increasingly, chrome-free pre-treatments. Have you considered this? Could this be a next step? Maybe the phrase ‘physical pre-treatment’ is a better phrase. I am not going to insist, I leave it to the editor – just something to consider going forward.
References
The references are well presented.
Round 2
Reviewer 1 Report
The author carefully revised the manuscript according to the reviewers' suggestions, and it seems that the current quality of the manuscript has been greatly improved. The author is suggested to redesign the cross page chart in the manuscript.
Author Response
The authors greatly appreciate the reviewer's time and comments. Thanks to the reviewers, the manuscript improved significantly. Regarding the last request made by the reviewer, figure 8 was re-arranged and has now a soother transition within 2 pages.